# Postnatal Growth and Development of the Rumen: Integrating Physiological and Molecular Insights

**DOI:** 10.3390/biology13040269

**Published:** 2024-04-18

**Authors:** Binod Pokhrel, Honglin Jiang

**Affiliations:** School of Animal Sciences, Virginia Tech, Blacksburg, VA 24061, USA; drbinod@vt.edu

**Keywords:** cattle, feed, IGF-I, rumen, volatile fatty acid

## Abstract

**Simple Summary:**

The rumen is a large organ unique to cattle and other ruminants and is essential for the ingestion, digestion, and absorption of large quantities of plant-based food. Cattle and likely all ruminants, however, are not born with a voluminous, functional rumen. There is remarkable postnatal growth and development in the rumen. This review summarizes the major postnatal changes in the structure and function of the bovine rumen, the major factors causing these changes, and, more importantly, the biological mechanisms underlying these changes.

**Abstract:**

The rumen plays an essential role in the physiology and production of agriculturally important ruminants such as cattle. Functions of the rumen include fermentation, absorption, metabolism, and protection. Cattle are, however, not born with a functional rumen, and the rumen undergoes considerable changes in size, histology, physiology, and transcriptome from birth to adulthood. In this review, we discuss these changes in detail, the factors that affect these changes, and the potential molecular and cellular mechanisms that mediate these changes. The introduction of solid feed to the rumen is essential for rumen growth and functional development in post-weaning calves. Increasing evidence suggests that solid feed stimulates rumen growth and functional development through butyric acid and other volatile fatty acids (VFAs) produced by microbial fermentation of feed in the rumen and that VFAs stimulate rumen growth and functional development through hormones such as insulin and insulin-like growth factor I (IGF-I) or through direct actions on energy production, chromatin modification, and gene expression. Given the role of the rumen in ruminant physiology and performance, it is important to further study the cellular, molecular, genomic, and epigenomic mechanisms that control rumen growth and development in postnatal ruminants. A better understanding of these mechanisms could lead to the development of novel strategies to enhance the growth and development of the rumen and thereby the productivity and health of cattle and other agriculturally important ruminants.

## 1. Introduction

Ruminants possess a complex stomach comprised of four compartments: rumen, reticulum, omasum, and abomasum. Among these, the rumen stands as the largest compartment in adult ruminants [1,2]. The rumen has an oval shape, and the adult rumen is internally divided into different sacs by the inflection of the muscular ruminal walls called ruminal pillars [2,3]. The inner surface of the rumen is covered with numerous finger-like projections called rumen papillae (Figure 1), which play an important role in nutrient absorption and metabolism [1,2]. The rumen plays an essential role in ruminant physiology and production. However, ruminants are not born with a large, functional rumen. The rumen grows considerably in mass and size after weaning [4]. The postnatal development of the bovine rumen can be divided into three phases: the pre-ruminant phase, which spans the initial two to three weeks when the calf relies solely on milk or milk replacer; the transitional phase, commencing once the calf begins to ingest solid feed until weaning; and the ruminant phase, occurring post-weaning when the rumen attains full maturation, and the calf uses volatile fatty acids (VFAs), the major products of fermentation, as its primary energy source [5]. Many of the functions of the rumen such as digestion, absorption, and metabolism are gained after weaning [3,5,6,7]. The development of the rumen from the pre-ruminant phase to the ruminant phase is controlled by a complex interplay of factors and mediators. These factors and mediators include solid feed, microbes, VFAs, hormones, and genes. Much research has been conducted to understand how the rumen develops during the postnatal stage. This review aims to summarize the postnatal changes in the bovine rumen, the factors affecting postnatal rumen growth and development, and the biological mechanisms underlying these effects.

## 2. Histology of the Rumen

Histologically, the rumen wall is composed of tunica mucosa, lamina propria, tunica submucosa, tunica muscularis, and tunica serosa, from inside to outside [4] (Figure 2A). Tunica mucosa is the innermost part of the rumen, is non-glandular, and has a superficial stratified squamous epithelium. The rumen epithelium consists of four cellular layers or strata: stratum basale, stratum spinosum, stratum granulosum, and stratum corneum, from base to apex (Figure 2B). Cells in these layers differ in morphology and function [4,8]. The cells in the basal layer (stratum basale) are columnar and play a role in nutrient absorption, ketogenesis, energy metabolism, and secretion of immunoglobulins [4,9]. The cells in the spinous layer (stratum spinosum) and granular layer (stratum granulosum) are multilayered and have the presence of tight junctions, adherens junctions, and desmosomes, which provide strong adhesion, resist mechanical stress, and maintain structural integrity [4,10,11]. Neither the stratum corneum nor the stratum basale expresses junctional proteins and hence has no barrier function [12]. The cells of the stratum granulosum lie perpendicular to the cells of the stratum spinosum [13]. The cells in the outermost layer (stratum corneum) of the rumen epithelium are highly keratinized and devoid of nuclei, and they serve as a protective physical barrier against rough, fibrous ingesta, and potentially toxic compounds [10,13]. It is believed that the keratinized cells of the stratum corneum are derived from the cells of the stratum granulosum. During this transition, granular cells lose their organelles, including the nuclei, ribosomes, mitochondria, and Golgi apparatuses, and as such, they cannot proliferate. Subsequently, all cellular filaments along with keratohyalin and endoplasmic reticulum protein complexes are displaced toward the cell periphery. As a result, a flattened horny cell with condensed content is formed. This horny cell is enveloped by the thickened plasma membrane, and during this transformation, only keratinizing epithelia filaments are retained, which subsequently get converted into a cohesive and protective keratinized stratum corneum [14].

The lamina propria fuses with the submucosa and supports, nourishes, and protects the mucosal layer [4]. There is an extension of the lamina propria and submucosa into the papillae [15]. These fused propria-submucosa layers lie beneath the mucosa and contain blood vessels, nerves, and lymphatic vessels, providing structural support, facilitating nutrient and oxygen delivery, and regulating physiological and immune functions [4,10]. The tunica muscularis consists of the longitudinal and circular smooth muscular layers responsible for the movement and contraction of the rumen, promoting effective mixing and fermentation of feed in the rumen [15]. Recently, telocytes have been identified as a novel structural component within and between the longitudinal and circular smooth muscles of the rumen, and they function to support the muscular layer and facilitate intercellular signaling, either directly or indirectly, through the telocyte network [16]. The outermost layer of the rumen is the tunica serosa, a thin layer of smooth tissue that functions to minimize the friction of the rumen with the surrounding organs [4,15].

## 3. Functions of the Rumen

### 3.1. Digestion

The rumen itself does not produce enzymes for the breakdown of plants and other feed eaten by a ruminant; however, it provides a habitat for various microbes, which include bacteria, protozoa, archaea, and different kinds of fungi that can break down complex plant and feed materials into readily available products for utilization [17,18,19,20]. These microbes can be categorized as cellulolytic, amylolytic, and proteolytic, responsible for degrading the cellulose, starch, and protein components of the feed, respectively [19]. Bacteria and protozoa are responsible for most of the digestion in the rumen, with bacteria accounting for 80% and protozoa 20% of the digestion [19,21].

Microbial digestion of complex plant polysaccharides produces short-chain fatty acids (SCFAs), which are also known as volatile fatty acids (VFAs), gases, and heat [19,22,23]. Major VFAs produced by rumen fermentation include acetic acid, propionic acid, and butyric acid [3,22]. The typical ratio of these three VFAs in the rumen ranges from 75:15:10 to 40:40:20, depending on dietary composition [22]. Major gases produced in the rumen include methane, ammonia, and carbon dioxide. Microbial digestion of complex polysaccharides in the rumen also yields lactic acid, formic acid, succinic acid, and other short-chain monocarboxylic acids [3].

Rumen microbes can convert esterified plant lipids to unsaturated fatty acids, which are further converted to saturated fatty acids by biohydrogenation [17,24]. Proteases and peptidases from rumen microbes diligently degrade feed proteins into peptides and amino acids, which are then incorporated either into microbial proteins or deaminated to form VFAs [25]. Microbial proteins are eventually digested and absorbed by the abomasum and the small intestines, accounting for 80% of total absorbable proteins in the ruminants [17,25]. Besides digestion, rumen bacteria can synthesize vitamin B complex and vitamin K [15,26]. In addition, the rumen epithelium can absorb ammonia and metabolize it to glutamic acid [27]. The rumen is also involved in waste elimination by moving methane, carbon dioxide, hydrogen, ammonia, and some other end products of rumen fermentation and undigested materials to the hindgut [28]. In addition, the rumen is involved in converting the potentially toxic products of rumen fermentation into metabolites that have less toxicity [29].

Ruminants have a special ability to regurgitate their ingesta from the reticulum to the esophagus and finally to the mouth for re-mastication, also known as cud chewing. This allows them to decrease the particle size and more efficiently digest the feed. It is believed that cud chewing also helps the ruminants provide saliva to maintain optimum rumen pH, increase the concentration of rumen microbes, and maintain the motility of the rumen [30].

### 3.2. Absorption of VFAs

The VFAs are the most important products of rumen fermentation. A dairy cow produces more than 100 moles of VFAs a day in the rumen [31,32]. Nearly 90% of total VFAs produced in the rumen are absorbed by the rumen epithelium [22]. The remaining VFAs are absorbed by the reticulum, omasum, abomasum, and to some extent, the small intestines [33]. The rate of absorption is directly proportional to the increasing chain length of fatty acids; thus, the rate of absorption of butyric acid is higher than that of propionic or acetic acid [22,33]. The number and size of the papillae are directly proportional to the capacity of the rumen to absorb VFAs, as they increase the surface area of the rumen epithelium [34]. Three sequential processes are involved in the absorption and transport of VFAs from the rumen (Figure 3). Firstly, equilibration of VFAs occurs between the rumen and the apical surface of the rumen epithelium. Subsequently, VFAs are absorbed into the rumen epithelium. Finally, the capillaries in the rumen epithelium remove VFAs from the basal side of the rumen epithelium into the blood [32,35].

Due to the high concentration of VFAs in the rumen, a concentration gradient of VFAs exists between the rumen and the bloodstream. This concentration gradient and the lipophilic nature of the rumen epithelium allow for the passive diffusion of non-ionized VFAs from the rumen into the rumen epithelium [18,36,37]. Once the non-ionized VFAs cross the cell membrane, they dissociate into anions and H^+^, which is the major source of hydrogen ions that leave the rumen epithelium (Figure 3). These hydrogen ions are recycled via a process called sodium/hydrogen exchange. Simultaneously, a bicarbonate ion exchange process occurs, which facilitates the absorption of ionized VFAs from the rumen into the rumen epithelium [18,31]. Bicarbonate ions are produced by the rumen epithelium in a considerable amount [38,39]. This process is important for the absorption of less lipophilic VFAs (e.g., acetic acid) into the rumen epithelium and finally into the bloodstream [38]. After the VFAs enter the rumen epithelial cells, an extensive metabolism of them (Figure 3, discussed in more detail later) occurs [40]. The metabolites of VFAs are absorbed into the bloodstream through a carrier-mediated process facilitated by downregulated in adenoma (DRA, also known as SLC26A3 or CLCA), pendrin anion exchanger 1 (PAT1, also known as SLC26A6), and monocarboxylate transporter 1 (MCT1, also known as SLC16A1) in exchange for chloride and bicarbonate ions [38,39,41,42,43].

### 3.3. Metabolism of VFAs

According to the older theory, approximately 90% of butyric acid, 50% of propionic acid, and 30% of acetic acid absorbed from the rumen are metabolized in the rumen epithelium to provide nearly 70% of the energy needs of a ruminant [22,40]. However, more recent work suggests that the extent of metabolism of acetic acid and propionic acid, particularly the latter, in the rumen epithelium might have been overestimated [40]. Acetic acid is the least metabolized VFA in the rumen epithelium, and partly because of this, it accounts for more than 90% of total VFAs in the arterial blood [22].

In the rumen epithelium, butyric acid and acetic acid are metabolized to the ketone bodies acetoacetic acid (AcAc) and β-hydroxybutyric acid (BHB) through ketogenesis (Figure 3) [4,44,45]. The 3-hydroxy-3-methylglutaryl-coA synthase, which is a rate-limiting enzyme in ketogenesis, is highly expressed in the adult but absent in the young rumen epithelium [46]. The rumen epithelium exhibits greater ketogenic activity with butyric acid than does the liver, which is the major site of ketogenesis in non-ruminant animals [45,47]. Ketone bodies derived from butyric acid and acetic acid are then used to generate energy in extrahepatic tissues, including the heart, skeletal muscle, adipose tissue, kidney, and mammary gland [40,48].

A trace of butyric acid, which is neither metabolized by the rumen epithelium nor by the liver, enters the adipose tissue or the mammary gland from the bloodstream, and there, is rapidly oxidized for lipogenesis [22]. In the rumen epithelium, a small proportion (about 5%) of propionic acid is converted to lactic acid and pyruvic acid [4,49] (Figure 3). The remaining propionic acid is used by hepatic tissue as a substrate for gluconeogenesis [3]. Most of the acetic acid absorbed from the rumen is not metabolized by the rumen epithelium or the liver and is instead utilized by peripheral tissues [22,50,51]. In the adipose tissue and mammary gland, acetic acid is a major substrate for *de novo* synthesis of fatty acids [52].

### 3.4. Barrier Function

The rumen epithelium is subject to challenges from rumen microbes, toxins produced by rumen microbes, changes in the ruminal pH, and many other potential adversities. The multi-layered rumen epithelium functions to efficiently maintain the separation between the ruminal lumen and blood circulation. This separation is achieved through tightly controlled selective absorption mechanisms within cells and intercellular junctions between cells that restrict the paracellular passage of ruminal pathogens and other insults into the rumen epithelium [38,53]. Besides the intercellular junctions, the immune cells within and beneath the rumen epithelium provide an “immune barrier” that protects the rumen epithelium from pernicious stimuli [54,55].

Rumen epithelial integrity and barrier function are achieved mainly through the intercellular junctions in stratum granulosum and stratum spinosum, which include tight junctions (TJ), adherens junctions (AJ), and desmosomes [56,57]. Tight junctions provide barrier functions, intercellular communication, cell adhesion, and selective permeability; adherens junctions and desmosomes provide strong cell adhesion and mechanical anchoring between the cells [58,59]. The keratinized layer present at the upper tip of the rumen papillae also protects the rumen against abrasion caused by feed particles and changes in rumen pH [55].

Various tight junction proteins have been identified in epithelial cells of different species, including claudins, occludin, cingulin, tricellulin, junctional adhesion molecule, and MarvelD3, which form semipermeable transmembrane along with the adapter protein and zona occludens 1, 2, and 3 [57,60,61,62]. Zona occludens link tight junction transmembrane proteins to actin cytoskeleton and help maintain the structure and functioning of tight junctions [58]. In rumen tissue and rumen epithelial cells, researchers have identified the protein presence of claudins 1, 4, and 7, and occludin [53]. Transcriptomic analyses have revealed the mRNA expression of claudins 1, 3, 4, 7, 12, and 23, cingulin, junctional adhesion molecules 2 and 3, occludin, and zona occludens 1 and 3 in rumen epithelial cells [7,63].

At low rumen pH, tight junctions in the rumen epithelium may lose their integrity or function, allowing harmful microbes to enter the bloodstream [56,64]. To combat this, activation of toll-like receptors by rumen microbes and their products, such as lipopolysaccharides (LPS), induces the production of cytokines, which trigger immune cells such as CD4+ T cells to resist the noxious stimuli [65,66] and the epithelial cells to increase the expression of zona occludens 1 and 2 [67]. Probiotics such as *Lactobacillus plantarum* MB452 also enhance the barrier function of the rumen epithelium by increasing the expression of tight junction genes [68]. Heat stress affects the rumen epithelial barrier negatively to some extent but does not completely disrupt it, due to the upregulation of heat shock proteins [69]. In essence, the strength of the physical barrier of the rumen is intricately linked with the microbes and immune cells in the rumen.

## 4. Prenatal Development of the Rumen

During bovine gastrointestinal development, which commences around the 29th day of gestation, the caudal region of the esophagus undergoes enlargement, leading to the formation of the definitive stomach compartments, although debates persist regarding the precise origins of these compartments, with evidence leaning toward development from a fusiform primordium [70,71]. By the 6th week of gestation, the rumen undergoes significant expansion and migrates to the left cephalic region, accompanied by the formation of blind sacs and spirally twisting vestibule, the narrow caudal portion of the rumen [70,71]. By around the 58th day of gestation, the dorsal and ventral blind sacs of the rumen expand, forming small evaginations of the main rumen, which later assume their definitive caudal position with a transverse notch, known as the caudal groove of the rumen, between them. During this period before birth, the caudal part of the vestibule develops into atrium ventriculi, while the cranial region merges with the dorsal sac of the rumen in adult animals [71]. At the end of the first trimester, the rumen and reticulum occupy the room between the 7th thoracic vertebrae and the 4th lumbar vertebrae. They extend consistently between the 3rd trimester and the birth to eventually occupy the abdominal cavity between the 9th thoracic and the 3rd lumbar vertebrae [72]. This period also witnesses a surge in cellular proliferation and differentiation within the rumen epithelium, resulting in a marked increase in cell layers [71,73]. Fully developed conical papillae, while still fused with the epithelium, are observable by the end of the pregnancy. By this point, the rumen epithelium already comprises the four layers (Figure 2) [73,74,75].

## 5. Postnatal Development of the Rumen

The rumen of a newborn calf is underdeveloped and not functional, and the digestive functions of the stomach are primarily performed by the abomasum, the largest part of the stomach at birth [4,6]. In pre-weaning ruminants, the ingested colostrum, milk, or milk replacer bypasses the rumen, reticulum, and omasum and goes directly from the esophagus to the abomasum through the esophageal groove [4,6,76]. The esophageal groove, also known as the reticular groove, is formed by the muscular fold of the reticulum and is a reflex response triggered by suckling [77]. The rumen epithelium of a newborn calf lacks keratinization and ketogenic activity, and the rumen of a newborn calf lacks an established microbial community [4]. The primary energy source for the neonatal rumen epithelium is derived from nutrients absorbed in the small intestine. Glucose, along with fatty acids, is considered a key energy substrate for the immature rumen epithelial tissue, especially before active fermentation begins in the rumen [78]. During postnatal life, the rumen undergoes significant changes, including increased muscle mass, epithelial growth, visible papillae formation, and colonization of microbial communities [20]. The length and width of rumen papillae increase with ages from birth to one year of age in cattle (Figure 4). The size of rumen papillae and the thickness of the rumen wall are often used as indicators of the degree of rumen development [79]. As the rumen matures, it gains the ability to metabolize VFAs and use VFAs instead of glucose as the major source of energy [80].

As a result of postnatal development, the proportions of the reticulorumen, omasum, and abomasum in the entire stomach undergo significant changes, shifting from 38%, 13%, and 49% at birth to 61%, 13%, and 25% at 8 weeks of age, and reaching 67%, 18%, and 15% at 12–16 weeks of age [76,81,82].

**Figure 4 biology-13-00269-f004:**
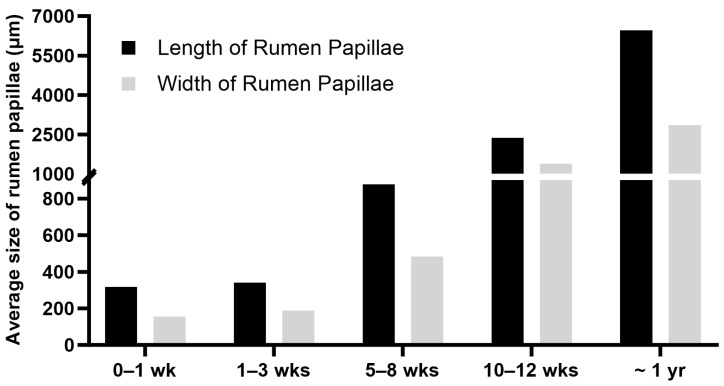
The size of rumen papillae in different age groups of cattle. Data obtained, grouped, and interpreted from these publications [13,83,84,85,86,87,88,89,90].

## 6. Factors Affecting the Postnatal Development of the Rumen

### 6.1. Diet

When fed exclusively milk, calves had limited rumen development; however, increased intake of solid feed accelerated rumen development in calves [91]. Rumen papillae grew in weaned calves but not in sucking and milk-continued calves [92]. Calves weaned earlier with higher solid feed intake showed more advanced rumen epithelial, papillary, and muscular growth in the rumen [6,93,94]. Lambs raised on a milk replacer alone showed limited metabolic maturation, while those weaned onto solid feed demonstrated increased VFA concentrations in the rumen and larger rumen papillae [91]. The proliferation rate of cells in the stratum basale was higher in calves fed milk replacer combined with a starter diet than in calves fed milk replacer alone [94]. All these data support the stimulatory effect of solid feed on rumen development in calves.

As mentioned earlier, when milk or milk replacer is drunk by a calf, it bypasses the rumen through the esophageal groove and directly enters the abomasum. Direct infusion of milk into the rumen promoted the growth of rumen papillae in calves, probably due to the availability of substrate for fermentation in the rumen [93]. However, it has also been reported that the introduction of liquid feed into the rumen within the first two weeks of birth can cause defective fermentation and, as a result, digestive disturbances such as ruminal acidosis and bloat occur, which can sometimes be fatal [77,95].

When inert substances such as nylon bristles, plastic tubes, and wood shavings, which are incapable of undergoing fermentation, were directly inserted into the rumen, it resulted in effective muscular growth in the rumen wall but no significant growth of rumen papillae [6,93]. The surface area of rumen papillae increased in a dose-dependent manner when dairy cows were fed different doses of rumen-fermentable organic matter during lactation and dry periods [96]. These findings suggest that the stimulatory effect of solid feed on rumen papillary growth is primarily attributed to the presence of active fermentation in the rumen, while the effect of solid feed on the growth of rumen muscle is probably due to the physical form of diet activating the stretch receptors in the muscular wall of the rumen [76,97,98].

### 6.2. Volatile Fatty Acids

As calves consume solid feed, rumen fermentation begins, and, as a result, rumen concentrations of VFAs rise. It has been widely believed that VFAs, particularly, butyric acid, mediate the effect of solid feed intake on rumen development [76,97,99]. Supplementation of sodium butyrate in calf starter promoted rumen development in newborn calves [85]. Intraruminal infusion of sodium butyrate increased the length of rumen papillae, the thickness of the stratum corneum, and the overall size of the rumen epithelium [100]. Sheep with rapid infusion of butyric acid showed greater mitotic activity in the rumen epithelium than sheep with slower infusion of butyric acid [101]. The expression of genes responsible for apoptosis, Caspase-3, and Bax, were significantly reduced in lambs fed sodium butyrate [100]. Volatile fatty acids stimulate rumen epithelial growth by increasing cell proliferation while reducing apoptosis [102,103].

VFAs influence the proliferation and apoptosis of rumen epithelial cells through various mechanisms. Dietary supplementation of ketone bodies, particularly β-hydroxybutyric acid (BHB), enhanced the growth of rumen epithelium [104], suggesting that butyric acid stimulates the growth of rumen epithelium through its metabolite BHB. Butyric acid and propionic acid might stimulate rumen papillary proliferation also by increasing blood flow through the rumen wall [76,105]. Butyrate can function as a histone deacetylase inhibitor to increase histone acetylation and thereby gene expression. There was a correlation between the production of acetic acid and butyric acid and the expressions of genes including MAPK1, PIK3CB, TNFSF1, ITGA6, SNAI2, SAV1, and DLG in rumen epithelium, all of which are related to cell growth [100]. Butyric acid has been shown to increase the expression of CCND1, CDK4, and PPARA, genes that promote cell cycle progression and lipid metabolism [97,106,107]. Thus, butyric acid might stimulate rumen epithelial growth by directly affecting the expression of genes involved in cell proliferation and metabolism through its histone deacetylase inhibitory activity.

### 6.3. Rumen Microbiota

As discussed above, the growth of rumen epithelium in postnatal ruminants is primarily driven by active fermentation and the production of VFAs within the rumen. Rumen fermentation and production of VFAs are facilitated by a diverse array of microflora consisting of bacteria, fungi, protozoa, and archaea, which colonize the rumen from maternal contact, environmental exposure, and feed intake [1,23,108]. These microorganisms perform fibrolytic, amylolytic, cellulolytic, and proteolytic activities [109]. The establishment of rumen microbiota is influenced by both host genetics and external environmental factors. The colonization of the rumen by microbiota can be observed as early as the first day of birth, transitioning from aerobic to facultative anaerobic and finally to strictly anaerobic bacteria as the animal matures [110,111].

The diverse microbial consortium operates cooperatively, fostering a conducive environment for fermentation and digestion, thereby influencing rumen growth. A correlation exists between ruminal VFA production and bacterial population [112]. Additionally, there is a correlation between the relative abundance of microbial communities and the expression of genes involved in various metabolic functions and molecular processes in rumen tissue [87]. Various taxa present in the rumen are responsible for the production of different enzymes that mediate the degradation of complex dietary materials and plant polysaccharides such as starch to glucose and finally VFAs [113]. Microflora in the rumen also interact with the host’s immune system, modulating various pathways to strengthen it [54,114]. Furthermore, the interaction between rumen microflora and the host contributes significantly to the regulation of the rumen’s barrier function and pH levels [54]. Some rumen bacteria have been found to digest keratinized epithelial cells, which promotes the growth of new cells in the rumen epithelium [115].

### 6.4. Hormones

Hormones and growth factors have significant effects on the growth and body composition of ruminants, and it would be unwise to disregard their involvement in rumen development and maturation. Insulin has long been suspected to mediate the stimulatory effect of VFAs on the proliferation of rumen epithelial cells in vivo [116]. This notion is supported by multiple lines of evidence. The promoting effect of butyric acid and propionic acid on rumen epithelial growth was associated with improved insulin sensitivity [117]. Infusion of insulin significantly stimulated cell proliferation in the rumen epithelium [116]. Treatment of primary rumen epithelial cells with insulin arrested the inhibitory effect of butyric acid on cell proliferation in vitro and increased the DNA synthesis of the cultured cells in a dose-dependent manner [118]. Nevertheless, not all studies support the role of insulin in mediating the effect of VFAs on rumen growth. For example, Wang et al. found that supplementation of isobutyric acid stimulated rumen development but was associated with decreased blood concentration of insulin in calves [119].

Calves weaned earlier (meaning transitioning to solid feed earlier) showed significantly more growth of the rumen and had higher concentrations of plasma insulin-like growth factor I (IGF-I) than calves weaned later [120]. Similar results were demonstrated in sheep [121]. The stimulatory effect of high-concentrate diets on the growth of rumen papillae was associated with increased systemic IGF-I concentration and increased binding of IGF-I to rumen epithelial IGF-I receptors in goats [121]. Supplementation of isobutyric acid increased rumen development and ketogenesis in calves in a dose-dependent manner, and these increases were accompanied by increased blood concentrations of IGF-I [119]. IGF-I stimulated the proliferation of rumen epithelial cells in vitro [121,122]. These observations suggest that intake of solid feed or butyric acid might stimulate rumen growth by circulating IGF-I [123]. However, short-term intraruminal infusion of butyric acid in castrated bulls increased the size of rumen papillae without increasing the concentration of plasma IGF-I [124], arguing against the role of IGF-I in mediating the effect of solid feed intake or VFAs on the growth of rumen epithelium.

Leptin is a hormone predominantly produced by fat and plays a central role in regulating appetite and energy balance [125]. Interestingly, leptin mRNA was expressed at elevated levels in the rumen and other stomach chambers in calves, indicating a potential involvement of leptin in the initiation of rumen development [126]. Ghrelin, a peptide hormone, is synthesized in the stomach and regulates growth hormone release, appetite, gastrointestinal motility, nutrient absorption, and energy balance in monogastric animals [127,128]. In lambs, ghrelin mRNA is abundantly expressed in the rumen [128], and this suggests a possible paracrine or autocrine action of ghrelin in the rumen. However, the role of ghrelin in postnatal rumen growth is not supported by the data indicating that rumen expression of ghrelin was not different between calves and adult cattle [126].

Epidermal growth factor (EGF) is a growth factor known for its function in regulating epithelial cell proliferation, morphogenesis, and tight junctions in the intestines [97,129]. Proliferation of rumen epithelial cells was stimulated by EGF in vitro [130]. EGF increased mRNA expression of tight junction proteins claudin 1 and occludin in the rumen [129]. These results support the potential role of EGF in regulating epithelial cell proliferation and tight junctions in the rumen.

Transcriptomic analyses suggested that transforming growth factor-β might be another hormone mediating the growth of rumen epithelium from pre-weaned to post-weaned calves [131]. Similarly, integrated analysis of differentially expressed long non-coding RNAs and mRNAs revealed thyroid hormone as another candidate hormone that mediates rumen development in postnatal animals [132,133].

### 6.5. Weaning

Weaning refers to the transition of young ruminants from dependency on maternal milk or milk replacer to consuming solid feed. Most calves start to consume solid feed after two weeks of age, and the intake of solid feed increases in an inverse relationship with the intake of milk [5,134]. Natural weaning starts when milk alone is insufficient to supply the daily requirements of the calves. This process is accompanied by a gradual reduction in the maternal–filial bond and rejection of dam to nurse the calves, forcing the calves to search for other sources of food [135].

While calves are naturally weaned at the age of six to nine months, it has become a common practice in cattle production to separate calves from dams earlier, as early as at the age of 6 weeks [136,137,138]. One of the benefits of early weaning is early maturation of the rumen [137,138,139,140]. However, as discussed earlier, early weaning could increase the risk of calves suffering from acidosis and other digestive disorders in later life. The development of the rumen following weaning is primarily attributable to the increased availability of substrate for fermentation, leading to the production of VFAs within the rumen. The pace of weaning also influences the development of the rumen. A recent study found that calves weaned gradually showed greater feed intake, higher body weight gain, increased immune functions, higher serum concentrations of β-hydroxybutyric acid, whereas calves weaned abruptly showed higher serum concentrations of non-esterified fatty acids and higher expression of inflammation-related genes such as IL-1β, IL-6, TNF-α, NF-κB, and ICAM in the liver or jejunum, indicating a more functional rumen in the former than the latter [141,142]. Weaning also changes the expression of genes related to rumen epithelial cell proliferation, molecular transport, and energy metabolism [138]. It is advocated that calves are gradually weaned over a period of two to three weeks at around three months of life for smooth transitioning from pre-ruminants to ruminants [138,140,143]. Interested readers can refer to previous reviews by Khan et al., Kertz et al., and Ghaffari and Kertz for recommended forms and amounts of solid feed intake for calves [134,144,145].

### 6.6. Other Factors

Feeding pasteurized colostrum and milk to neonatal calves has a significant positive impact on their overall health status; calves fed pasteurized colostrum and milk products developed fewer or no respiratory and digestive pathologies compared to those fed unpasteurized products [146,147]. However, feeding pasteurized milk and milk products has been associated with lower rumen mass compared to calves fed whole milk or milk replacers during the weaning phase (60 to 180 days of age) [148]. Feeding pasteurized colostrum and milk products to calves had conflicting effects on the composition of the ruminal microbiome [147].

Probiotics, prebiotics, synbiotics, growth promoters, and digestive enzymes fed to ruminants can affect rumen development indirectly by affecting the digestion of feed and production of VFAs in the rumen [76,84,149]. Supplementing calves’ diet with microbes or probiotics such as *Bacillus licheniformis*, *Saccharomyces cerevisiae*, *Bacillus subtilis natto*, *Lactobacillus plantarum*, and live or hydrolyzed yeast has the potential to enhance rumen development by influencing the microbial community and digestion in the rumen [150,151,152]. The addition of inulin to whole milk of calves increased the length and width of the rumen papillae [84]. Antibiotics added to feed to increase the productivity of animals could modulate the development and function of the rumen [153,154]. Including pectinase in the diet of lambs increased the production of major VFAs in the rumen, suggesting its effect on rumen development or function in young ruminants [155].

The development of the rumen is influenced by various pathological conditions, including digestive pathologies. Experimental infection of pre-weaned dairy calves with salmonellosis, a major causative agent of digestive problems in the calves, has been shown to negatively affect the growth of ruminal papillae [156]. Similarly, digestive issues induced by the higher intake of lactose, either in milk or in milk replacers in young calves, result in delayed or lesser growth of the rumen compared to those without digestive problems [157]. In adult steers, gastrointestinal toxicosis or acidosis has been observed to cause serious degeneration of the ruminal epithelium; such pathologies can alter the acid-base balance of the rumen, disrupt its microbial ecosystem, and impair its absorptive capacity, ultimately inhibiting rumen development [34,158,159].

## 7. Genes Involved in Rumen Growth and Development

Structural and functional development of the rumen can be eventually attributed to changes in gene expression in the rumen. The expression of 6 transcription factors (OTX1, SOX21, HOXC8, SOX2, TP63, and PPARG) and 16 ruminant-specific conserved non-coding elements (RSCNEs) contributes significantly to differentiating the rumen from the esophagus [160]. Functioning as enhancers or silencers, RSCNEs contribute to pre- and post-transcriptional regulation, modulation of genes associated with morphological and physiological traits, and chromatin architecture, and play a crucial role in shaping the distinctive features of the rumen [132,161,162]. Zhang et al. identified 6048 differentially expressed genes (DEGs) in the rumen among neonatal, young, and adult cattle [163]. In experiments conducted by other groups, 2048 rumen DEGs were identified between 1-day- and 56-day-old calves, and 871 rumen DEGs were found between suckling and weaned calves [80,164]. Micro RNAs (miRNAs) are small noncoding RNAs that play an important role in post-transcriptional regulation of gene expression. A total of 132 miRNAs were differentially expressed in the rumen between the pre- and post-weaning calves [161].

It is safe to say that the development and functional maturation of the rumen relies on the coordinated expression of various genes involved in diverse molecular functions and biological processes. Genes such as CCNB1, CCNB2, IGF1, IGF2, HMGCL, BDH1, ACAT1, and CREBP could play important roles in driving rumen development because they are known to function in cell proliferation and fatty acid metabolism, and they were expressed at higher levels in the adult cattle than calf rumen [163]. Genes encoding the IGF binding proteins IGFBP2, IGFBP3, and IGFBP6 were expressed at higher levels in the rumen of suckling and milk-continued calves than that of weaned and solid feed-fed calves [92]. Furthermore, the expression levels of IGFBP2, IGFBP3, and IGFBP6 negatively correlated with the growth of rumen papillae [92]. IGFBPs in general function to block the binding of IGF-I to its receptor. These gene expression data support the hypothesis that reduced expression of IGFBP genes in the rumen contributes to solid feed-induced growth of rumen papillae in post-weaning animals, perhaps by increasing the mitogenic action of IGF-I on rumen epithelial cells. Interestingly, in at least one study, IGFBP5 was found to be upregulated during the growth of rumen papillae [164]. Perhaps IGFBP5 has the ability to regulate the growth of rumen papillae independent of IGF-I.

Members of the ruminant-specific PRD-SPRRII gene family were expressed at significantly higher levels in the mature than immature rumen [165]. Genes such as ACTG2, ILK, and KIF4A, which are involved in cell skeleton regulation, cell adhesions, and cell organization, were among the genes expressed at higher levels in adult rather than young ruminants [97,166,167]. Similarly, FABP7, FABP3, ILK, PDGFA, HMGCS2, and AKR1C1 were differentially expressed between the immature and mature rumen [87,168]. Rumen expression of FABP3 was downregulated in weaned calves compared to pre-weaning calves [169]. The upregulation of FABP3 is linked with inhibition of cell growth and proliferation, suggesting a possible role for the FABP3 gene in inducing the proliferation of rumen epithelial cells in post-weaning calves [168,169]. MCT1 (SLC16A1), PAT1 (SLC26A6), ACAT1, HMGCS2, HMGCL, ACAA1, ACAA2, CA4, PPARA, PPARG, SLC26A3, and other members of the SLC family are highly expressed in the adult rumen [80,112,170]. These genes participate in rumen development and functional maturation by facilitating the transportation of VFAs from the ruminal lumen to the rumen epithelial cells and to the bloodstream, or by facilitating beta-oxidation and ketogenesis of VFAs in the rumen epithelium [164,169].

Members of the claudin (CLDN) family, CLDN1, CLDN4, and CLDN7, are highly expressed in the rumen [53], and they may be of prime importance in maintaining the integrity of the rumen epithelium because they encode tight junction proteins [171]. Similarly, genes coding for tight junction-associated proteins MarvelD3, occludin, and tricellulin likely play a role in maintaining the structural integrity and epithelial barrier function of the rumen too [61]. Genes DMRT2, WDR66, COL7A1, EVPL, KRT14, F2RL1, TMPRSS13, and TMPRSS11A participate in the cell junction biological process, are upregulated in the adult rumen [80] and, therefore, may also contribute to the barrier function of the adult rumen.

CEBPB, S100A9, CCL20, CXCL8, TNF, NFKBIA, LBP, PGLYRP2, MUC1, SOCS1, SOCS3, and CCDC3 activate the innate immunity in the rumen [80]. BPIFA1, part of the BPI-like protein family, plays a role in the innate immunity of the epithelial mucosa [97]. These genes are likely involved in both the development and health of the rumen.

## 8. Conclusions

The proper development of the rumen, particularly the rumen epithelium and papillae, is crucial for the physiology and production of cattle and other ruminants. While the rumen of newborn calves shares anatomical and histological similarities with that of adult cattle, the rumen undergoes significant development during postnatal life, especially after weaning. The rumen encompasses remarkable physical and functional changes in postnatal animals. Physically, the rumen grows in mass and size; functionally, the rumen gains the ability to ferment feed into VFAs (because of colonization of microbes and intake of solid feed), absorb VFAs, metabolize VFAs, and protect against potential insults from the fermentation. Among the various factors influencing postnatal rumen growth and functional maturation, a solid diet stands out as the most significant. Introducing solid feed to the rumen is essential for its growth and functional development in young ruminants. Solid feed stimulates rumen growth and functional development through VFAs, especially butyric acid, produced by rumen fermentation. Butyric acid and other VFAs stimulate rumen growth and functional development through both direct and indirect mechanisms. In direct mechanisms, butyric acid and other VFAs stimulate cell proliferation and metabolism in rumen epithelial cells by altering the expression of genes critical for these processes. In indirect mechanisms, butyric acid and other VFAs stimulate cell proliferation, and metabolism by increasing the concentration and/or action of circulating insulin-like growth factor-I (IGF-I), insulin, and perhaps other hormones and growth factors on the rumen. While these mechanisms have been proposed repeatedly in the literature, they await validation through carefully designed studies. A deeper understanding of how diet and other factors affect rumen development, growth, and function at the cellular, molecular, genomic, and epigenomic levels may help devise novel or more effective management practices to enhance rumen development and functional maturation in cattle and other agriculturally important ruminants.

## Figures and Tables

**Figure 1 biology-13-00269-f001:**
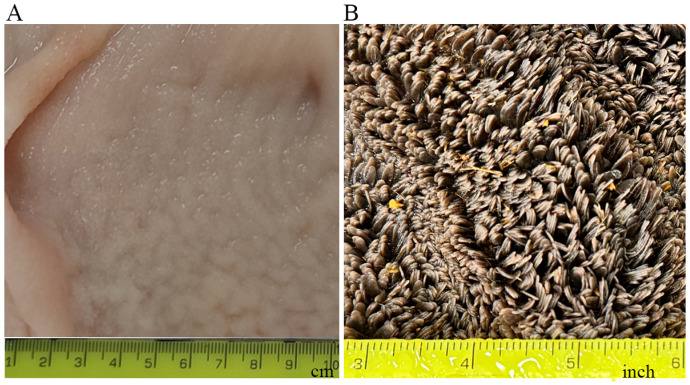
Macroscopic view of the internal surface of the rumen. (**A**) The rumen of a newborn calf; (**B**) The rumen of an adult steer. Note that compared to the newborn rumen, the inner surface of the adult rumen is covered with large rumen papillae.

**Figure 2 biology-13-00269-f002:**
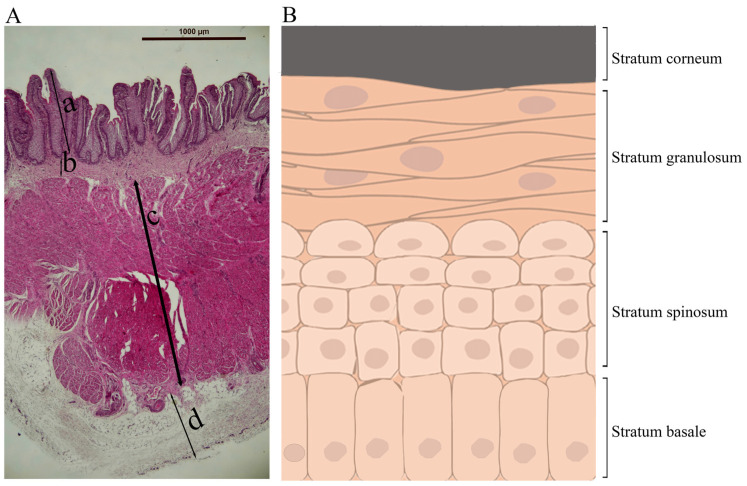
(**A**) Microscopic view of the rumen wall. The section was cut from the rumen of a newborn calf and stained with hematoxylin and eosin. a: rumen papilla; b: mucosal and submucosal layer; c: muscular layer; d: serosal layer. (**B**) Schematic representation of different cellular layers of the rumen epithelium.

**Figure 3 biology-13-00269-f003:**
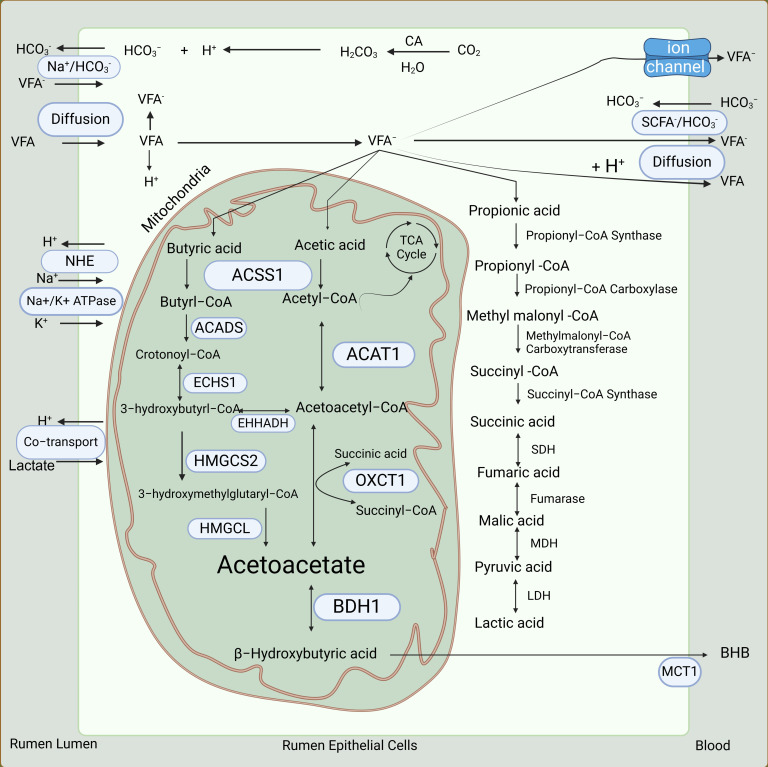
Schematic representation of absorption and metabolism of volatile fatty acids. CA: carbonic anhydrase; VFA: volatile fatty acids; ACSS1: acyl-CoA synthetase short-chain family member 1; ACADS: acyl-CoA dehydrogenase short-chain; ECHS1: enoyl-CoA hydratase, short chain 1; ACAT1: acetyl-CoA acetyltransferase 1; EHHADH: enoyl-CoA hydratase and 3-hydroxyacyl CoA dehydrogenase; HMGCS2: 3-hydroxy-3-methylglutaryl-CoA synthase 2; HMGCL: 3-hydroxy-3-methylglutaryl-CoA lyase; OXCT1: 3-oxoacid CoA-transferase 1; BDH1: 3-hydroxybutyrate dehydrogenase 1; SDH: succinate dehydrogenase; MDH: malate dehydrogenase; LDH: lactate dehydrogenase.

## Data Availability

Not applicable.

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
