# Peer review of "Postnatal Growth and Development of the Rumen: Integrating Physiological and Molecular Insights"

_biology, 2024, doi:10.3390/biology13040269_

Round 1

Reviewer 1 Report

Comments and Suggestions for Authors

The study gives a full review of postnatal growth and development of the rumen from physiological and molecular insights.  The author gives us a detailed description of the development of rumen histology and the function of the rumen in ruminants, and emphasizes the importance of solid feed in the development and growth of rumen. 

There is a flawed description in line 363. Please modify the description of Wang and colleagues to Wang et al. The same as line 425.

Reviewer 2 Report

Comments and Suggestions for Authors

The present review explains "to summarize the postnatal changes in the bovine rumen, the factors affecting postnatal rumen growth and development, and the biological mechanisms underlying these effects". This research provides interesting information. However, some changes need to be made before final publication

General comments: I recommend that the review be oriented from the general to the particular, mentioning the importance of an early transition, factors that affect this transition, how the type of production system influences it, tools that can improve "Development of the Rumen", etc.

Introduction

General comments: I recommend increasing this section. Mention the factors related to "Postnatal Growth of the Rumen".

Line 28.- delete a “u” in “uunderstanding.

Line 240.- in the section "Prenatal Development of the Rumen" I recommend to expand this section with more information.

Line 251.- in the section "Postnatal Development of the Rumen", mention the importance of the transition from a liquid to a solid diet and the mechanisms that affect this transition. Can digestive pathologies (e.g. diarrhea) affect this development? Also, can the use of pasteurized colostrum and milk contribute to a good development of the calves?

Lines 254-256.- "Milk or milk replacer consumed by a calf bypasses the rumen through the ruminoreticular groove and enters directly into the abomasum for digestion [4,73,74]". What factors favor this "bypasses the rumen" (ruminoreticular groove).

Line 264.- “The length and width of rumen papillae increase with ages from birth to one year of age in cattle (Figure 4). The size of rumen papillae and the thickness of rumen wall are often used as indicators of the degree of rumen development [76]”. What conditions diminish this development?

Line 279.- “however, increased intake of solid feed accelerated rumen development in calves [88]”. What is the recommended consumption of solid feed and what type?

Line 297.- “These findings suggest that the stimulatory effect of solid feed on rumen papillary growth is primarily attributed to the presence of active fermentation in the rumen, while the effect of solid feed on the growth of rumen muscle is probably due to the physical form of diet activating the stretch receptors in the muscular wall of the rumen [74,93,94]”. However, it is important to mention that the entry of liquid feed (milk), when the rumen is not fully developed, can cause problems of poor fermentation causing digestive pathologies in the calf.

Line 328.- “Rumen Microbiota” Are there treatments based on probiotics and prebiotics during this stage in calves, which can favor their development?

Lines 405-406.- “One of the benefits of early weaning is early maturation of the rumen [131-133].” What do you mean by the term "early weaning" 45 or 60 days? In addition, I suggest to increase this section to mention which are the indicators for a good "Weaning", times and how we can use this tool to improve the "Development of the rumen".

Reviewer 3 Report

Comments and Suggestions for Authors

Pokhrel et al., summarized the factors that contributes to the postnatal development of rumen. The paper is well-written and the organization of the manuscript is good. But I feel most topics in the review are well-known. There is not that many new concepts. Besides, instead of only summarize what other people have done, you should also have your perspectives on the implications of these studies and what might be the future work.  

If Figure 1 and Figure 2 are not developed by your own. Please indicate where they are from.
